# Mentors matter: Association of mentors with project success in the Apache Software Foundation Incubator

**Curtis Atkisson** ⓘ *

School of Public Policy, University of Massachusetts, Amherst, Massachusetts, United States of America

* catkisson@umass.edu

## Abstract

Mentoring has been a subject of study for 50 years. Most studies of mentoring programs evaluate the effect of the program on the participants but do not evaluate if different mentors have different effects on mentees. Open-source software (OSS) is software with a license that allows it to be freely used by other people. Such software has become foundational to the world economy. However, many OSS projects get abandoned by their creators. Various nonprofit organizations have arisen to help OSS projects become sustainable. One of the key services offered by many of these nonprofit organizations is a mentorship program where experienced OSS developers advise nascent projects on how to achieve sustainability. We use data from the Apache Software Foundation Incubator program where 303 mentors have mentored 286 projects, with most mentoring more than one project, to address this question: Is who a project has as a mentor associated with variation in project success? Who a project has as a mentor accounts for 45% of the variation in project outcomes, with some mentors being associated with positive and some with negative outcomes. These mentors could offer insights into how to improve the mentoring program. This result also demonstrates, more broadly, that the nature of specific mentoring relationships may be important to understanding how mentors impact outcomes in other mentoring programs.

## Introduction

Most mentor-mentee relationships are intense interpersonal relationships where individuals interact over long periods of time [1]. These relationships have been shown to impact an individual's rating of the program, likelihood of continuing in a career, and even overall career success [2, 3]. Due to the intense nature of these relationships, however, research typically focuses on mentee outcomes, and little is known about how individual mentors vary in the impact they have on mentees.

Open-source software (OSS) underlies much of the software we use on a regular basis and contributes substantially to the world economy [4]. The production of open-source software requires private, typically otherwise highly-compensated, individuals to donate their time and energy to produce a public good (something, by definition, from which they cannot exclude

**Data Availability Statement:** All relevant data are within the article and its Supporting Information files.

**Funding:** This material is based upon work supported by the National Science Foundation (nsf.

gov) under GCR grants no. 2020900 and 2020751 awarded to BB, CS, SF, and VF. The funders had no role in study design, data collection and analysis, decision to publish, or preparation of the manuscript.

**Competing interests:** The authors have declared that no competing interests exist.

others, and which is not depleted by someone else's use)—a classic collective action problem in that costly individual contributions are required to a public good. While many open-source projects fail or become abandoned [5], many are successful or sustained well enough to become vital parts of high-value enterprise infrastructure, such as in the servers running Google's search services, the software underlying your banking software, and the infrastructure required for the big-data revolution.

Whether or not it is emergent or designed, the way that OSS is produced leads it to be well-characterized as a commons-based peer production system [6, 7]. In such systems, people work together, cooperatively, with potentially little (though highly variable) hierarchy to produce artifacts that can be used freely by anyone able to access them (e.g., someone must know how to use the repository in which code is housed and how to make the code run). By its nature, software is non-rivalrous, in that one person running an instance of a piece of software typically does not prevent someone else from running another instance of that software. By design, through decisions such as hosting software on a public site and the license it is under, OSS may be non-excludable, meaning that potentially no one can be prevented from using the resource. This makes OSS a classic public good [8]. Such public goods may have collective action problems associated with them: individuals (including organizational actors) may benefit from the maintenance of the public good but have no incentive to contribute to maintaining the public good [9].

In the last 30 years, supporters of OSS created nonprofit organizations that act as umbrella organizations for OSS projects, providing support (e.g., legal) in exchange for some limited control over the project (e.g., the project will be branded as part of the foundation), many with the stated aim of overcoming the collective action problem and helping to produce more OSS as a public good. In order to be beneficial to solving the collective action problem, these organizations must be adding some value to the OSS public good. One primary way that they add value to projects is by guiding nascent projects to sustainability, typically through an incubator program. A cornerstone of these incubator programs is mentoring, whereby experienced OSS developers will advise the project to help it achieve sustainability.

We use long-term data to test whether mentors and the mentoring program in general contribute to project outcomes in the Apache Software Foundation (ASF) Incubator. In this program, hundreds of mentors have advised hundreds of projects, many mentors have advised more than one project, and projects have several mentors. This results in analytical challenges, but also provides a rare opportunity to answer the fundamental question of if the mentor program has an effect on project outcomes, and if specific mentors give their projects a better chance of success. Who mentors a project accounts for 45% of the variation ($R^2$) in project graduation. ASF is good at enrolling projects they expect to graduate (base expected probability of graduation of 77%), providing a greater scope for mentors to make a project more likely to not graduate than to graduate, though we identify groups of mentors with positive and negative effects on graduation probability. This establishes that who one has as a mentor is associated with project outcomes. This demonstrates that who serves as a mentor for a project matters to the potential success of that project and provides scope to examine how mentors contribute to the success of OSS projects.

This paper adds to the literature on both mentoring and OSS production. It contributes to the literature on mentoring by examining whether who a project has a mentor impacts success in the program, something that is difficult to evaluate in many mentoring contexts. This work adds to the literature on OSS production by showing that an aspect of many Incubator programs, mentoring, may be improved by studying the effect of various mentors on project success and sharing strategies used by those mentors. The main research question addressed is if there is individual variation in the relationship between mentors and project success. The

paper precedes by a presentation of background on OSS, OSS umbrella organization, OSS Incubators, and mentoring; followed by the methods used; the results found; and a discussion of those results in the context of both mentoring and OSS production.

# Background

## Open-source software

Open-source software (OSS) is software that is released under a copyright license allowing non-copyright holders to access, use, and (sometimes) modify the software [10]. There are a variety of licenses ranging from the relatively "restrictive" General Public License (GPL), which requires that any product that uses code released under that license be released under equivalent license terms, to so-called "permissive" licenses such as the Apache license, which allows for the code to be used in any way without requiring resultant products to be released under equivalent licenses [10]. These licenses allow developers to freely build off of the work of others, providing the potential for efficiency gains in producing new software. The OSS operating system Linux—often seen as the poster child of OSS—powers approximately 2% of desktop computers, at least 50% of servers, and nearly 100% of supercomputers across the globe [11, 12]. More practically, many of our daily interactions on the internet, from collaboratively edited documents to banking software, critically rely on OSS. A recent study estimates that OSS contributes between 65 and 95 billion Euros to the EU economy [4].

There is no one way to produce OSS, though the nature of the process means that there are broad similarities across OSS projects. Typically, an individual or small group of individuals will identify a need, either in the market or, more usually, for themselves, develop a solution to a minimum standard (typically marked as "alpha" or "beta"), distribute the software in some way (typically just by announcing it on their personal communication channels), and wait for people to begin using the software [13]. These users are encouraged to contribute to the software, first by identifying bugs in the software and submitting feature requests. For many projects, the goal is to transition users to contributors: people who write code to fix bugs or add features. At this point, the creator(s) will review all changes before including them in the software. If a contributor contributes enough to become trusted by the creator(s), they can be elevated to what different projects variously call "committer" or "maintainer" status, providing them the ability to approve code and incorporate it directly into the software. There are many skills involved in this process that the typical programmer may not have (i.e., marketing, management, etc.). Deciding on a license and ensuring that all code adheres to that license is an important legal consideration, an additional skill that a typical programmer may not have.

## OSS umbrella organizations

Over the last 30 years, supporters of OSS have created nonprofit organizations (herein referred to as foundations, as many of them style themselves) that act as umbrella organizations for open-source software projects, providing support (e.g., legal, technical) in exchange for some limited control over the project (e.g., the project will be branded as part of the nonprofit organization). The first of this type of organization was the Free Software Foundation, which was established in 1985 by the originator of the "copyleft" copyright protections (i.e., GPL), Richard Stallman. Initially, funds managed by the Free Software Foundation were used to pay people to produce free software, though that transitioned in the mid-1990's so that funds were being used to address legal and structural issues in the production of OSS. Since then, such foundations have proliferated, with more than 100 offering a range of services to OSS projects [14]. Izquierdo and Cabot [14] found that most foundations assist in community development, software development, and the sponsoring/promotion of specific OSS projects. On the other

hand, foundations require money and effort to stay afloat. Some foundations require that their members pay a fee, some solicit donations from corporations that use the software sponsored by the foundation, and others allow companies to "pay to play" to get a seat at the table leading an OSS project.

Foundations, therefore, act as an additional layer trying to address the collective action problem. By mobilizing development effort and centralizing skills needed by OSS projects that the typical OSS developer may not have, they may be a boon to the OSS commons, resulting in the creation of more, higher quality, or more sustainable freely available software. However, as these organizations often require time invested by OSS developers into the organization, there is a possibility that they result in less, lower quality, or less sustainable freely available software. While maybe not explicitly acknowledging this, such organizations usually offer a range of services that will help, rather than hinder, the production of OSS [14]. One of the primary ways that foundations positively contribute to the OSS commons is through incubator programs that match OSS projects with experienced OSS developers in formal mentorship programs.

## Incubator programs and graduation and retirement

Different foundations have different expectations regarding the projects under their auspices, and a foundation's Incubator program is typically how the foundation ensures that those expectations are met. The aim of most Incubator programs, then, is two-fold: to help new OSS projects become sustainable and to ensure that the project is run in a way the foundation thinks is good. It is common to describe projects that have made it through an Incubator as graduated and those which have failed to make it through the Incubator as retired or archived.

The foundation studied here is the Apache Software Foundation (ASF). ASF was founded on the heels of the Apache Web Server, which quickly rose to prominence as the most popular web server only a few years after being released as OSS. Since then, ASF has grown by accepting new projects that will eventually become "top-level projects": projects which have been deemed sufficiently advanced to run their entire project and use ASF branding without supervision. The only way that a new project enters ASF, with very few exceptions, to become a top-level project is through the Incubator. Projects enter the Incubator after a vote is taken on their proposal, after which they get a team of mentors. While in the Incubator, projects are expected to release software that complies with the Apache license (one of the "permissive" licenses described above) and adhere to The Apache Way, a set of principles that ASF believes are the best way of making OSS. Once projects have demonstrated their ability to release code that conforms to the Apache license and follow The Apache Way, a vote for their graduation will take place, after which they become a top-level project.

## Mentoring

One of the services some OSS foundations provide is project mentoring. Mentoring, generally, as an interpersonal relationship between an experienced individual and novice, became a topic of study in the 1970's and 80's [1–3]. Almost universally, studies of mentors or mentor programs show a positive effect on some outcome of interest such as perception of the relationship, sentiment regarding the position the mentee was in, likelihood to continue in a career track, or impact on career [1–3, 15–19]. Nearly all of these studies focus on how having a mentor impacts the mentee's perception of their experience, though a few examine the effect of mentoring on some real-world outcome.

Software systems can be large-scale, complex artifacts that are typically built up over time and by many people, which makes learning a new system difficult, leading some to liken it to being an immigrant in a foreign country [20]. Early work on using mentoring programs as a

tool to introduce newcomers to a software system focused on teams in corporations [21]. More recently, however, the focus of the study of mentoring in software has included OSS [22–24]. It is well-recognized that one of the most difficult parts of running an open-source project is giving potential contributors their first task to work on—one that is simple enough to accomplish yet complex enough to keep them engaged [23]. As many of the papers on mentoring in OSS projects are from the perspective of software engineering, they are interested in designing tools and automatic recommendation systems for mentors [22, 23], and less in evaluating mentoring programs.

The ASF Incubator adheres to The Apache Way, which, amongst its tenets, emphasizes free-choice and volunteerism as key mechanisms to produce high-quality OSS. As a part of this, very little, if any, of the interactions between mentors and the projects they mentor is prescribed. Additionally, all of these mentors are programmers first, who are expected to help projects move towards graduation by sharing their experience. This provides a large scope for individual variation in mentoring strategies to impact project success. Indeed, there is not even a prescribed size of the mentorship team, though three is the minimum and experienced mentors often say that five mentors is ideal. Projects are free to identify and recruit their own mentors, and to have as many as they think is appropriate

## Research questions

The studies described above, both within and outside OSS, highlight that being enrolled in a mentoring program is good for mentees as it may increase their perception of the experience or even increase their career prospects. None of the studies, however, evaluate the variation among mentors to see if there are some mentors that lead to better outcomes than others.

This brings us to our focal research question: *Does who a project has as a mentor matter for the OSS project's graduation or retirement from the ASF Incubator*? Or, to rephrase this, is it possible to identify individual variation in outcomes associated with mentors?

To answer these questions, we focus on the Apache Software Foundation. Data from the ASF Incubator program allow us to assess how much of the variability in successfully graduating from the program is due to who someone has a mentor, and if there are some mentors that have positive and negative effects on graduation rates. The data used in this paper are unique because we have 303 mentors who have mentored 286 projects. Mentorship teams can be any size (ranging from one mentor for early projects in the Incubator to ten for one popular project) and there is no inherent ordering in mentor importance. This setup allows us to truly address the question of how much variation in success is associated with who a project has as a mentor and if specific mentors are associated with the probability of graduation from the ASF Incubator.

## Methods

The data are publicly available through the Apache Software Foundation (https://apache.org) and were created by scraping mentor data from the *projects page* (https://incubator.apache.org/projects/) and each *individual project's page* (e.g., https://incubator.apache.org/projects/amber.html). These are all the ASF Incubator OSS projects that have fully graduated or retired over the ASF lifetime. As the mentors on the projects page and on each individual project's page sometimes differ, for the purposes of this study the mentors for each project are the union of the set of mentors on the projects page and on each individual project's page. This leads to a dataset where each OSS incubator project is a case and whether or not each mentor was a mentor for that project is a variable. This type of dataset can be referred to as a "sparse" design matrix [25], where many of the values for each variable are zero (i.e., most mentors are not a mentor on a given project).

We predict graduating (vs. retiring) from the ASF Incubator by whether or not each mentor was a mentor for that project. In other words, we predict graduation with the fixed effect of each mentor as a binomial regression with a log link. Because our number of predictors was greater than the number of observations, and we anticipate that many mentors have little direct association with the graduation probability of projects they mentor, we preceded the binomial regression with a variable selection step using LASSO (least absolute shrinkage and selection operator; [25, 26]). This method allows us to reduce the effects included in the final regression. We measure the extent to which variation in the outcome is accounted for by the included effects through Tjur's Coefficient of Discrimination [27], which is a pseudo-$R^2$ measure that can be interpreted the same as "accounting for the variance in the outcome", and tells us the ability for a model to discriminate between binary outcomes. Together, these results allow us to characterize the importance of the mentorship program and individual variation in a mentor's association with outcomes.

A number of alternative modeling strategies were considered and rejected in favor of this method. A binomial model predicting graduation probability for each mentor individually was rejected because larger mentorship teams have a higher graduation rate, meaning those projects, in effect, get counted more often in the data set. A typical random effects analysis where the random effect is mentor would suffer from the same problem (projects with more mentors would be counted more often). Multiple Bayesian analyses were considered including a hierarchical parameter for fixed mentor effects (which failed because the Bayesian Fraction of Missing Information was low, indicating that the model was poorly specified) and a horseshoe prior (which suffered from sensitivity to prior specification). Despite mismatches of these methods to the data on first principles, all results from them agreed with the results presented below.

All analyses are conducted in R [28] using the package glmnet [29] to calculate the LASSO model. Code and data may be found in the S1 File.

## Results

### Descriptive statistics

The histogram of the distribution of the number of ASF Incubator projects ever mentored by a person shows that many (N = 128) have only ever mentored one project and 175 have mentored more than one project, with the majority of those mentoring two projects and it falling off steeply beyond that (Fig 1). On the opposite extreme, one person mentored 28 projects. Overall, the number of projects mentored has a mean of 3.4, a median of 2, and a standard deviation of 3.9 projects mentored. Mentors can mentor more than one project simultaneously, with prolific mentors often doing so.

Projects that enter the incubator tend to graduate. In this dataset of 264 incubated OSS projects, 204 have graduated and 60 have retired, leading to an empirical graduation rate (number of graduated/number of projects) of 77%. The distribution of graduation rate by mentors is in Fig 2, where black bars are the number of mentors for a given graduation rate who have mentored more than one project and white bars are the number of mentors who have mentored only one project (and for whom there can only be two possible outcomes). Most of the weight is at 100% graduation rate since most projects in the ASF Incubator graduate. Many mentors have only ever mentored one project, which may have retired, leading to a large number at 0% graduation rate. As two projects is the second most frequent number of projects a mentor has mentored, there is an increased number of mentors with a 50% graduation rate (i.e., one graduated and one retired project) relative to other values.

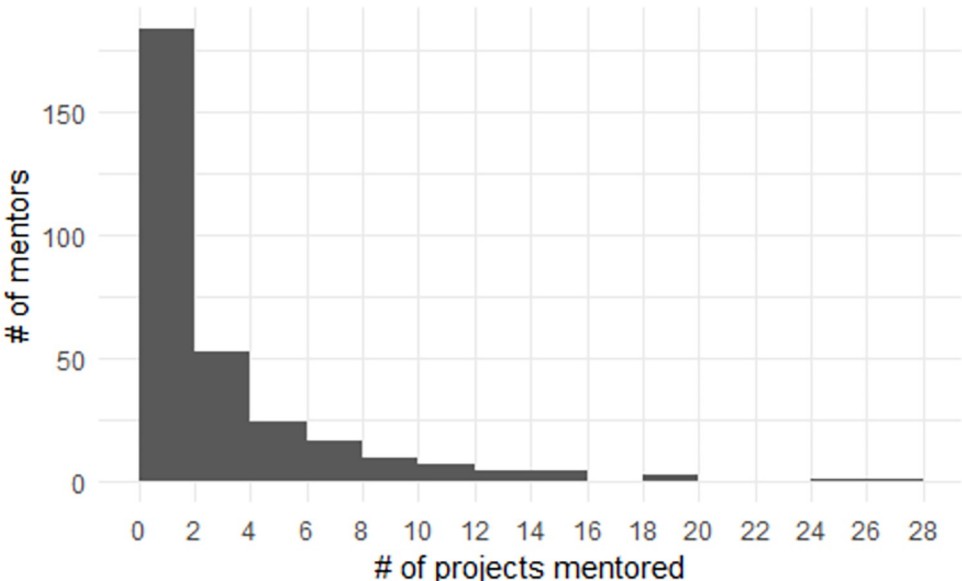

**Fig 1. Distribution of number of mentored projects.**

We can examine the graduation rate for mentors who have mentored a given number of projects (Fig 3). In this figure, the points give the average graduation probability for mentors who have mentored a given number of projects. The size of the points is the number of mentors who have mentored that number of projects; point size is continuous, though only intervals of 20 are shown in the legend. The lines around each point give the standard error of the mean graduation probability for the mentors who have mentored a given number of projects, giving some sense of the variability around the mean. This figure appears to show multiple trends. At the broadest level, the graduation probability tends to increase as we move left to right, indicating that mentors who have mentored more projects are associated with high

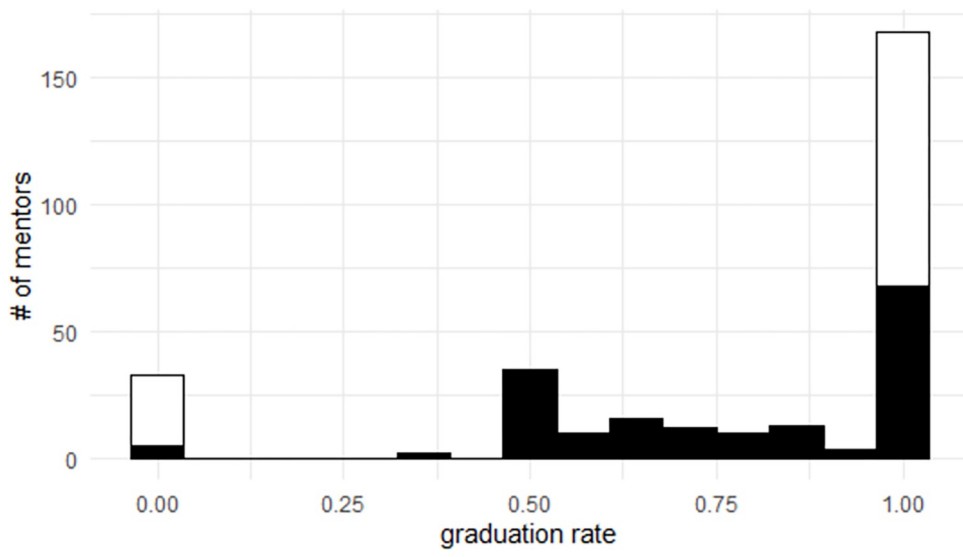

**Fig 2. Distribution of graduation rate by mentors.**

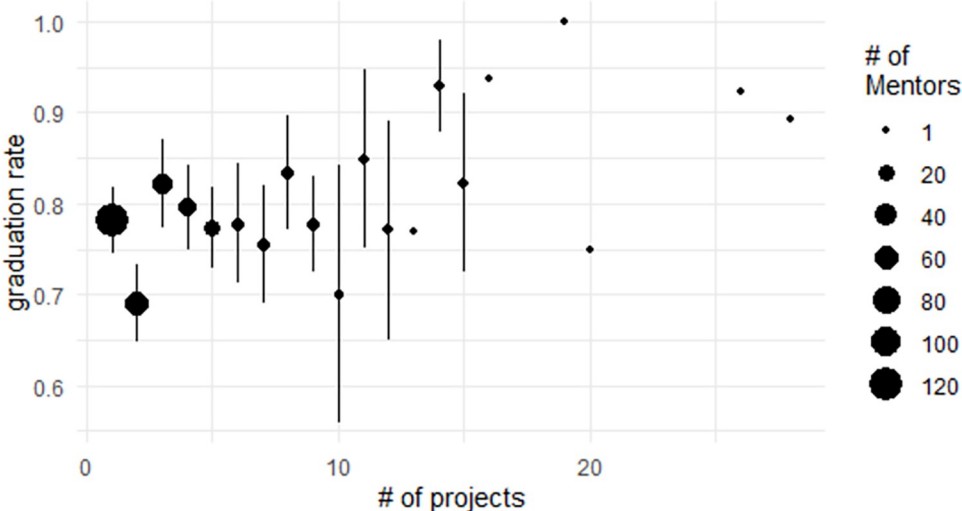

Fig 3. **Average graduation rate for mentors who have mentored a given number of projects.**

graduation rates. This broad trend, however, hides some important variation. Of particular interest, the graduation probability seems to decrease as mentors go from mentoring three to seven projects. As few mentors have mentored more than 10 projects, the standard errors increase at that level, with only single individuals having mentored any more than 15 projects.

As mentioned earlier, ASF incubator OSS projects can have more than one active mentor. Examining graduation rate from the project perspective, the broad trend is that having more mentors increases graduate rates (Fig 4). This figure gives the graduation rate for projects that have a certain number of mentors. As above, the size of the dot shows how many projects had that many mentors. The current expectation at the ASF Incubator is that projects will have at least three mentors, with Incubator management believing that three to five mentors is ideal (personal communication with the VP of the Incubator), leading to the large number of

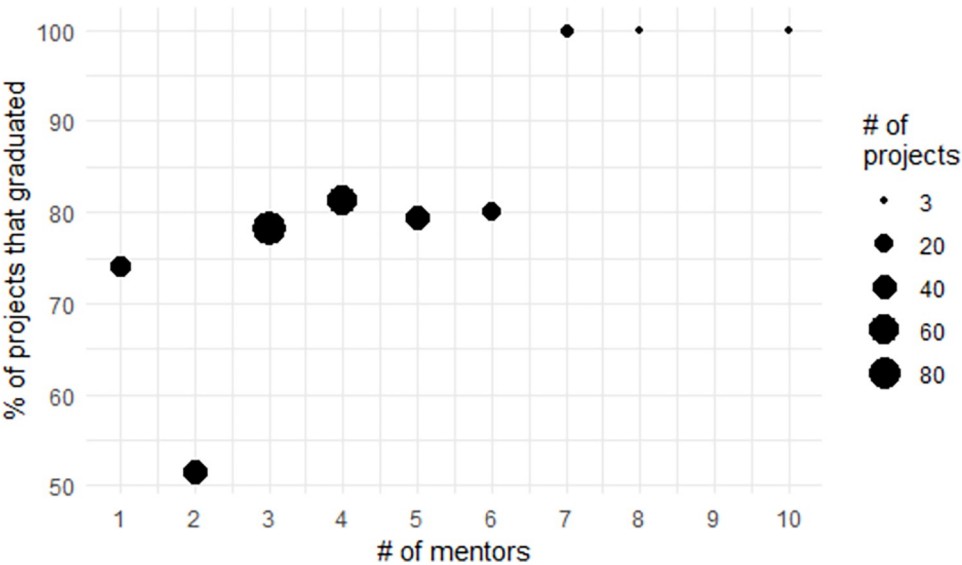

Fig 4. **Graduation rate for projects with a certain number of mentors.**

projects with that many mentors (N = 190). Within that range, the graduation probability is mostly stable, though with slight increases throughout the range relative to having three mentors. Of the 14 projects with seven or more mentors, all of them graduate. A binomial regression predicting graduation from the number of mentors gives a change in the predicted probability of graduating from one to 10 mentors of 33%. From this, it seems clear that having more mentors leads to a higher graduation rate. As such, we incorporate this variable into our statistical models going forward.

## Statistical modeling results

Here we follow the analytic steps described in the last part of the Methods section. The LASSO process is run including the effect of 303 mentors and the number of mentors a project has (for a total of 304 coefficients) on 286 projects. This results in 34 coefficients being retained (see Table 1). Since regularized estimates are biased to an unknown extent, only the direction and relative magnitude of the effect should be interpreted [26]. As such, the values in the Effect column are relative to the largest absolute value of the raw effects (-1.63). Of the mentor coefficients retained and estimated in the GLM, 30 have negative effects and 3 have positive effects. Of those with negative effects, 22 of them are from mentors who mentored only one project. In effect, this means that, given the other mentors on the team, the inclusion of this individual as a mentor lowered the chance of the project graduating. The largest effect is for a mentor who mentored three projects, all of which retired. The relative effect of the number of mentors on the project is 0.06 (raw effect 0.09), meaning that projects with larger mentorship teams have an increased chance of graduating. Tjur's coefficient of determination is 0.45, meaning that 45% of the variation in project outcome can be accounted for by knowing only who a project has as a mentor. The same analysis run on the subset of mentors who have mentored at least two projects retains all of the mentors from Table 1 and no additional mentors, lending further support to the idea that these mentors are meaningfully associated with graduation probability.

## Discussion

These results demonstrate (A) that who a project has as mentor and (B) how many mentors a project has is associated with graduation rate. Indeed, 45% of variation in graduation rate is accounted for just by knowing who a project has as their mentors. This demonstrates that substantial variation in graduation from the ASF Incubator is associated with the mentorship program. It also shows individual variation in mentors. In short, these results answer our earlier research question: *mentorship matters in determining whether a project graduates or retires from ASF's incubator program*.

   Despite that, it is the case that most mentors do not have a distinguishable effect on the graduation probability of projects that they mentor: only 33/303 of all mentors and 11/175 of mentors who have mentored at least two projects have effects that are distinguishable from zero. As most projects have at least three mentors, it is likely that the effects of mentors balance each other out, especially when people are first learning how to mentor. Of those mentors who do have an impact, only three of them have a positive effect. This is unsurprising as the proposal process to enter ASF is rigorous and projects get a lot of assistance to graduate, leading to a high empirical graduation rate. These mentors with a positive effect, however, clearly act in some ways that help projects move towards graduation.

   The trend shown in Fig 3 where the graduation probability decreases for mentors who have mentored three to six projects may have implications for the mentorship program in the ASF Incubator. This may indicate that mentors with experience, but only a small amount of experience, are being given more responsibility for the success of the project but may not yet be

**Table 1. Coefficient estimates from LASSO model.**

| Predictor | Effect | # projects | # graduated |
|---|---|---|---|
| Intercept | 0.57 | — | — |
| Mentor 1 | -0.01 | 1 | 0 |
| Mentor 2* | 0.03 | 12 | 12 |
| Mentor 3 | -0.13 | 1 | 0 |
| Mentor 4 | -0.18 | 1 | 0 |
| Mentor 5 | -0.25 | 1 | 0 |
| Mentor 6* | -0.80 | 2 | 0 |
| Mentor 7 | -0.29 | 1 | 0 |
| Mentor 8 | -0.31 | 1 | 0 |
| Mentor 9* | -0.74 | 2 | 0 |
| Mentor 10* | -0.05 | 6 | 3 |
| Mentor 11* | -0.86 | 2 | 0 |
| Mentor 12 | -0.35 | 1 | 0 |
| Mentor 13 | -0.35 | 1 | 0 |
| Mentor 14 | -0.01 | 1 | 0 |
| Mentor 15 | -0.13 | 1 | 0 |
| Mentor 16 | -0.25 | 1 | 0 |
| Mentor 17* | 0.20 | 19 | 19 |
| Mentor 18 | -0.01 | 1 | 0 |
| Mentor 19* | -0.30 | 12 | 6 |
| Mentor 20 | -0.19 | 1 | 0 |
| Mentor 21 | -0.19 | 1 | 0 |
| Mentor 22* | -0.77 | 2 | 0 |
| Mentor 23 | -0.37 | 1 | 0 |
| Mentor 24* | 0.08 | 14 | 14 |
| Mentor 25 | -0.31 | 1 | 0 |
| Mentor 26 | -0.31 | 1 | 0 |
| Mentor 27* | -1.00 | 3 | 0 |
| Mentor 28 | -0.02 | 1 | 0 |
| Mentor 29* | -0.17 | 3 | 1 |
| Mentor 30 | -0.20 | 1 | 0 |
| Mentor 31 | -0.02 | 1 | 0 |
| Mentor 32 | -0.37 | 1 | 0 |
| Mentor 33 | -0.43 | 1 | 0 |
| # of mentors | 0.06 | — | — |

* denotes that a mentor is retained in the analysis

Containing those who have mentored >1 project

experts in moving projects toward graduation. A recommendation to fall from this could be that the ASF Incubator should measure the performance of mentors and seek to ensure that the mentorship team for a project includes people at different levels of experience with mentorship: an experienced mentor to guide the project, an emerging mentor to gain more experience, and a novice mentor to introduce them to the program. While this may be difficult for an all-volunteer program like ASF to implement, it could increase the graduation probability of projects in the Incubator. This type of setup may also benefit other mentorship programs that use mentorship teams composed of mentors with varying levels of experience.

Even though most of the effects of individual mentors reported here are negative (i.e., having a specific mentor decreases graduation probability *below the population graduation probability*), this does not mean that the mentor program has a negative impact on mentors. Indeed, the ASF Incubator has a demanding proposal process for new projects, and the Incubator is good at enrolling projects that are likely to graduate (either because projects likely to graduate self-select in or the Incubator selects projects likely to graduate). As we do not test the counterfactual of projects with no mentoring, we cannot say whether the program itself is good for projects. Indeed, the process of joining the ASF requires mentors, making it impossible to graduate without a mentor. These results do, however, show that how a project is involved with the mentor program is associated with graduation probability.

These findings have implications for OSS production more broadly. OSS projects are complex artifacts that are difficult to begin working with, meaning that projects with more effective mentorship programs will successfully incorporate more contributors into their project, increasing their long-term sustainability. OSS projects also require a large number of skills that fall outside of the expertise of a typical software developer, including management, legal, and IT. The ability of creators and maintainers to manage the complex needs of the projects will impact their long-term sustainability. Advice from experienced managers of OSS projects on these aspects of the projects, which is what mentors do in the context of ASF, can help ease the transition of developers to become managers of complex projects. This highlights mentoring both of developers within a project and of managers of projects as key determinants of success in producing OSS.

These findings also have implications for mentoring programs more broadly. Many mentoring programs involve long-term interpersonal relationships, meaning that many mentors have few mentees, making evaluating the effect of certain mentors on real-world outcomes difficult. The dataset used here allows us to evaluate the performance of specific mentors. The finding that who one has as a mentor, not simply that they have a mentor, has a strong association with outcomes indicates that mentors should be evaluated for their effectiveness in mentoring programs more broadly.

## Threats to validity

This dataset includes all of the projects that have participated in the ASF Incubator and all of their mentors, decreasing threats to validity. Despite that, there are a couple complications that make interpretation of these results difficult. In essence, all the threats to validity deal with independent association of a variable of interest to the dependent variable. For instance, there may be a selection bias amongst some mentors towards mentoring projects more likely to graduate. This could arise if, for instance, graduating a project was a matter of prestige, and mentors were chasing prestige. Anecdotal evidence from ongoing qualitative work with mentors, however, indicates that is not the case. Furthermore, most mentors mentor just a couple projects, decreasing the chance that mentors are seeking prestige by mentoring. An additional possibility is that projects with larger code bases or more committers require more mentors and are also more likely to be successful. This could be investigated with further research. Even if this is the case, however, the mentors identified as having substantial impacts on project graduation would still have been identified as such. These threats to validity serve more to provide scope for future research than to threaten the conclusions of this research.

## Implications for future research

The structure of the ASF Incubator makes interpreting the association between specific mentors and graduation probability complicated. A number of processes could have led to this

association. Mentoring at the ASF Incubator is a volunteer effort. As such, some mentors may choose projects to mentor based on their perception of the likelihood that the project will graduate. There is also individual variation in domain-specific skills and interests that may lead some mentors to be more likely to work with a project that will end up graduating or retiring. For example, a developer who mainly works on big data software may be more likely to volunteer to mentor a big data project, and those projects may be more likely to graduate. Other domain-specific areas on which people may differ are the types of technology they use (e.g., programming languages) and skills that apply to the ASF Incubator, such as ensuring license compatibility or helping projects build community.

Natural Language Processing and the automated analysis of text provides a particularly exciting avenue through which we can further learn what makes for (un)successful mentoring. In open communication systems (such as at ASF) all communications are retained and publicly available. A classifier has been designed that categorizes emails as containing statements about how a project should be conducted and can label those from mentors [30]. The semantic content of those emails could be analyzed and related to project success, pointing towards types of mentoring that lead to project success

These reasons point to further research both within and outside the ASF Incubator. Conducting this same research in organizations where mentors are assigned and participation is required will help us understand how general these results are. Understanding how patterning in areas of interest/skill matches patterning in graduation will help further contextualize these results and examining mentoring programs in organizations with fewer dimensions on which skill and interest may differ will help generalize these results. Finally, data on specific strategies mentors use to guide projects will help us understand exactly how having a certain mentor leads to increased graduation probability.

## Conclusion

Despite these future research directions, the results shown here demonstrate that (1) 45% of variation in graduation may be attributable to the mentorship program and (2) that there is mentor-level patterning in graduation probability. This finding provides scope to investigate how some mentors may help projects have higher chances of graduating from the ASF Incubator and becoming sustainable in the long-term. These results also open the possibility that such mentor-level patterning in outcomes may be present in other mentoring programs.

This result applies more broadly than just to the ASF Incubator—also to mentoring programs of all kinds. In most mentor programs and assessments of those programs, only the impact of being a part of the mentoring program is investigated. This may include looking at mentees' reports of the relationship, measuring the change in their interest in an area after engaging in the program, or looking at the impact on mentees' careers. They do not, however, typically assess the impact that specific mentors have on those outcomes. This is unsurprising, because mentor relationships are usually involved and costly, leading most mentors in mentor programs outside of ASF to mentor only a limited number of mentees. Because mentoring at the ASF Incubator is done by teams, many mentors are able to mentor multiple projects, if not simultaneously then at least sequentially. These results demonstrate that mentor programs could benefit from being able to measure the effect of particular mentors on outcomes of interest and indicate that evaluators of such programs should strive to evaluate the effects of different mentors.

Open-source software is a valuable public good from which potential users are not excluded and the use of which does not prevent others from using it. The creation of OSS requires many skills that a typical software developer may not have, including marketing, management, and

legal skills. OSS umbrella organizations have arisen that strive to create more, higher quality, and/or more sustainable OSS by centralizing many of these skills and offering them to OSS projects as needed. One primary service these organizations offer is project incubation and mentoring. These results demonstrate that, for at least one such organization, the way projects interact with the mentoring program is associated with a substantial portion of variation in graduation from the Incubator program and point towards ways to improve the mentoring program. For this organization, the mentoring program seems to be an important area of interaction between the organization and OSS projects. To whatever extent this holds true generally, OSS umbrella organizations are providing a tremendous service towards the development of the OSS commons and a valuable public good.

## Supporting information

**S1 Data.**
(CSV)

**S1 File.**
(R)

## Acknowledgments

The author would like to acknowledge the contribution of Drs. Brenda Bushouse, Charlie Schweik, Vladimir Filkov, and Seth Frey; the entire Jumpstarting Successful Open-Source Software Projects With Evidence-Based Rules and Structures team; the Computational Institutional Analysis and Design lab at the University of Massachusetts—Amherst; and, especially, Likang Yin for helping initially scrape mentor data. Members and mentors at The Apache Software Foundation provided valuable information to contextualize these results. Feedback on previous versions of this paper was received at the International Association for the Study of the Commons—Knowledge Commons and ApacheCon@Home 2021 conferences. All mistakes that remain despite the best efforts of all those people are my own.

## Author Contributions

**Conceptualization:** Curtis Atkisson.

**Data curation:** Curtis Atkisson.

**Formal analysis:** Curtis Atkisson.

**Investigation:** Curtis Atkisson.

**Methodology:** Curtis Atkisson.

**Project administration:** Curtis Atkisson.

**Resources:** Curtis Atkisson.

**Software:** Curtis Atkisson.

**Supervision:** Curtis Atkisson.

**Validation:** Curtis Atkisson.

**Visualization:** Curtis Atkisson.

**Writing – original draft:** Curtis Atkisson.

**Writing – review & editing:** Curtis Atkisson.

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
