## [Decision Letter · Decision Letter 0]

14 Mar 2022

PONE-D-21-32481Mentors matter: Association of mentors with project success in the Apache Software Foundation IncubatorPLOS ONE

Dear Dr. Atkisson,

Thank you for submitting your manuscript to PLOS ONE. After careful consideration, we feel that it has merit but does not fully meet PLOS ONE’s publication criteria as it currently stands. Therefore, we invite you to submit a revised version of the manuscript that addresses the points raised during the review process.

 Both reviewers felt the paper I can proceed along the reviewing process. Though both emitted a Minor revision verdict, the sum of the two sets of suggestions (especially those by Reviewer 1) call for a significant revision effort on your side.

We look forward to receiving your revised manuscript.

Kind regards,

Maurizio Naldi

Academic Editor

PLOS ONE

Journal Requirements:

2. Please note that in order to use the direct billing option the corresponding author must be affiliated with the chosen institute. Please either amend your manuscript to change the affiliation or corresponding author, or email us at plosone@plos.org with a request to remove this option.

Reviewers' comments:

Reviewer's Responses to Questions

**Comments to the Author**

1. Is the manuscript technically sound, and do the data support the conclusions?

Reviewer #1: Yes

Reviewer #2: Yes

2. Has the statistical analysis been performed appropriately and rigorously? 

Reviewer #1: Yes

Reviewer #2: Yes

3. Have the authors made all data underlying the findings in their manuscript fully available?

Reviewer #1: Yes

Reviewer #2: Yes

4. Is the manuscript presented in an intelligible fashion and written in standard English?

Reviewer #1: Yes

Reviewer #2: Yes

5. Review Comments to the Author

Reviewer #1: The manuscript was mostly well-written and the work is compelling. This is important research but I believe the manuscript can be improved with some revisions. The following notes (with line/figure numbers) describe my requests. They mostly relate to accuracy and clarity in presentation of your work:

37 Be more explicit about what you mean by (and how it is) "a classic collective problem"

39-40 - Open Source is so ubiquitous, it is actually downplaying its reach to just refer to Google's search engine servers. It would even be helpful to cite some of the most prolific open source projects (currently).

44 It is a bit misleading to characterize all projects as having "little hierarchy" when the organization/collaboration hierarchy varies substantially

50-53 Similarly, the "public good" that "no one can be prevented from using" paints projects with too-wide a brush. Some licenses restrict usage (i.e. no commercial use) or place other limitations. The Background section is appropriate and more nuanced so even some curbing of the absolutes with "some" or even "many" would suffice.

96 Avoid scare quotes.

104-107 If these are documented descriptions of "committer" and "maintainer" roles somewhere in ASF documentation (or in another reputable source), please cite them.

198-199 "This brings us to our focal research question: Does mentorship matter for OSS project incubator

199 graduation or retirement?" could be stated more clearly since the project studied differences between mentors, not the difference between mentorship vs not. Instead, it seems you are really investigating the differences between individual mentors and/or the *quantity* of mentors for a project

376-377 This statement is doing a lot of hard work: "Though given the complexities of joining the ASF Incubator, it would seem impossible to graduate without a mentor." Please elaborate on those complexities with detail.

The outcomes are interesting and seem to be interpreted appropriately from the data. However, the discussion/implications need to go further. Some matters should be addressed, such as:

1) Are mentors helping on more than one project simultaneously, do they only mentor a new project when a previous one has graduated, or a mix of both?

2) How are mentors (and number of mentors) determined for a project?

3) Is there a relationship between number of mentors and the size/scope of the project? Or the number of committers?

4) Is there a relationship between the number of projects that mentors have taken on and their respective projects' graduation rates? A simple correlational test would be helpful

5) Now that there is some indication of individual mentors' association with graduation rate, what can (and can't) be extrapolated from that? What are some future studies that can be conducted to investigate what makes for (un)successful mentoring?

There have been some insinuations of threats to validity (such as pointing out selection bias in mentors choosing projects likely to graduate) but these threats to validity should be explicitly stated. It warrants having a dedicated (sub)section.

The figures are out of order, but I will assume that will be fixed for the camera-ready version.

Figure 1 should have more ticks (with labels) because it is hard to interpret whether each bar is "bucketing" more than one number or if there is something else going on

Figure 4 is the y-axis rate (i.e. rate for number of mentors is better than chance) or percentage (as labelled, which means <1% graduation rate for most). I suspect the former, but if that is the case, the axis label should be fixed.

Reviewer #2: The title, abstract, and keywords, accurately reflect the manuscript’s content.

In the introduction section, I suggest that the authors add a new paragraph addressing the research gap, introduce the focal research question, and add a paragraph presenting the manuscript structure/content.

The literature review is appropriate and addresses all the relevant subjects.

Methods seem to be adequate (please note that I am not an expert in statistics) and are well explained.

The same happens with the results and findings. However, the manuscript will benefit from a more elaborate discussion, comparing the obtained results with the prior literature.

Overall, the manuscript is well written and well structured, and the language is easy to understand and correct.

Minor issues:

- In the introduction, the acronym “ASF” (Apache Software Foundation) should be defined before the first use.

- The reference list should be revised since some references are incomplete (ex. ref. 22).

6. PLOS authors have the option to publish the peer review history of their article (what does this mean?). If published, this will include your full peer review and any attached files.

Reviewer #1: **Yes: **Kevin Buffardi

Reviewer #2: No

---

## [Author Response · Author response to Decision Letter 0]

26 Apr 2022

Editor Naldi,

Thank you for your handling of this paper. The reviewers you solicited provided great thoughts. Incorporating their comments has made the paper stronger. I respond here to the comments from reviewers. Hopefully this allows the paper to be published. Please let me know if you have any questions or comments.

Reviewer 1:

Lines 37, 39-40, 44, 50-53, 96, 104-107, 198-199 were dealt with as suggested by the reviewer

Line 376-377: The statement was clarified as just qualifying our results. As stressed above, we are not testing whether having a mentor matters, but whether specific mentors matter

1)

This is addressed in the revision on lines 277-278

2)

This is addressed in the revision on lines 203-206

3)

Unfortunately, these data do not exist in this dataset. This is a good avenue for further research. This had been added into the new “Threats to validity” subsection”

4)

This is shown in Figure 3 and addressed in the paragraph spanning from lines 296-308. A simple correlational test does show that there is a relationship, though (as noted in the prose), that simple result hides some important variation in the distribution of graduation rate by number of projects mentored. I believe that presenting the simple correlation would obfuscate the important underlying dynamics.

5)

This was previously address in the “Implications for future research”. This has been expanded slightly with a specific idea to investigate what makes for (un)successful mentoring (lines 426-432).

There have been some insinuations of threats to validity (such as pointing out selection bias in mentors choosing projects likely to graduate) but these threats to validity should be explicitly stated. It warrants having a dedicated (sub)section.

 A threats to validity subsection was created.

The figures are out of order, but I will assume that will be fixed for the camera-ready version.

 I believe that I submitted the Figures correctly labeled. Please let me know if I have erred. Thanks!

Figure 1 should have more ticks (with labels) because it is hard to interpret whether each bar is "bucketing" more than one number or if there is something else going on

 Ticks have been added for each bucket

Figure 4 is the y-axis rate (i.e. rate for number of mentors is better than chance) or percentage (as labelled, which means <1% graduation rate for most). I suspect the former, but if that is the case, the axis label should be fixed.

 The y-axis is % of projects with that many mentors that have graduated. Fig title has been changed to be consistent and axis label has been expanded

Reviewer 2:

In the introduction section, I suggest that the authors add a new paragraph addressing the research gap, introduce the focal research question, and add a paragraph presenting the manuscript structure/content.

 Added this paragraph lines 82-91

- In the introduction, the acronym “ASF” (Apache Software Foundation) should be defined before the first use.

Corrected

- The reference list should be revised since some references are incomplete (ex. ref. 22).

 Corrected

Thank you for your time!

Curtis Atkisson

---

## [Decision Letter · Decision Letter 1]

25 May 2022

PONE-D-21-32481R1Mentors matter: Association of mentors with project success in the Apache Software Foundation IncubatorPLOS ONE

Dear Dr. Atkisson,

Thank you for submitting your manuscript to PLOS ONE. After careful consideration, we feel that it has merit but does not fully meet PLOS ONE’s publication criteria as it currently stands. Therefore, we invite you to submit a revised version of the manuscript that addresses the points raised during the review process.

Though the paper has substanntially improved, one referee has observed some problem with the representation of results and the lack of support for some statement.==============================

We look forward to receiving your revised manuscript.

Kind regards,

Maurizio Naldi

Academic Editor

PLOS ONE

Reviewers' comments:

Reviewer's Responses to Questions

**Comments to the Author**

1. If the authors have adequately addressed your comments raised in a previous round of review and you feel that this manuscript is now acceptable for publication, you may indicate that here to bypass the “Comments to the Author” section, enter your conflict of interest statement in the “Confidential to Editor” section, and submit your "Accept" recommendation.

Reviewer #1: (No Response)

Reviewer #2: All comments have been addressed

2. Is the manuscript technically sound, and do the data support the conclusions?

Reviewer #1: Yes

Reviewer #2: Yes

3. Has the statistical analysis been performed appropriately and rigorously? 

Reviewer #1: Yes

Reviewer #2: Yes

4. Have the authors made all data underlying the findings in their manuscript fully available?

Reviewer #1: Yes

Reviewer #2: Yes

5. Is the manuscript presented in an intelligible fashion and written in standard English?

Reviewer #1: Yes

Reviewer #2: Yes

6. Review Comments to the Author

Reviewer #1: Thank you for your revisions, which have addressed most of my concerns. However, I see three issues still remaining:

1. The representation of Figure 4 is still incongruent with the other data presented. The y-axis suggests that the percentage of projects that graduated ranges from 0.5% to 1% (depending on number of mentors) and that the vast majority of projects are not represented in this chart. I suspect this is a problem of the scale label, where 0.5% should really be 50% and 1.0% should really be 100%. If that is the case, the scale should be fixed (or axis label corrected to clarify that it is the graduation *rate*, and not the *percentage*).

2. Even with more tick marks, Figure 1 is still difficult to interpret. The ticks fall in between (and not aligned with) each bar. Consequently, it is unclear at x=2 if the y-value is ~175 or ~50.

3. Lines 404-5 state: "Ongoing qualitative work with mentors, however, indicates that is not the case." There is no substantiation provided. If there are publications of this work, please cite them; if there are novel data, please present them (along with the methods). Otherwise, it should be restated that the authors believe it is not the case based on their anecdotal observations.

Reviewer #2: I am happy with the revision made by the author.

As I said in my previous review, the manuscript would benefit from a more elaborate discussion, comparing the obtained results with the prior literature. The author did not make this improvement, but the manuscript quality is sufficient for publication.

7. PLOS authors have the option to publish the peer review history of their article (what does this mean?). If published, this will include your full peer review and any attached files.

Reviewer #1: No

Reviewer #2: No

---

## [Author Response · Author response to Decision Letter 1]

8 Jul 2022

Editor Naldi,

Thank you for your handling of this paper. I have incorporated the suggestions of Reviewer 1 and took the opportunity allowed by Reviewer 2 to expand on the Discussion. Please let me know if you have any questions or comments.

1. The representation of Figure 4 is still incongruent with the other data presented. The y-axis suggests that the percentage of projects that graduated ranges from 0.5% to 1% (depending on number of mentors) and that the vast majority of projects are not represented in this chart. I suspect this is a problem of the scale label, where 0.5% should really be 50% and 1.0% should really be 100%. If that is the case, the scale should be fixed (or axis label corrected to clarify that it is the graduation *rate*, and not the *percentage*).

THIS HAS BEEN CORRECTED

2. Even with more tick marks, Figure 1 is still difficult to interpret. The ticks fall in between (and not aligned with) each bar. Consequently, it is unclear at x=2 if the y-value is ~175 or ~50.

INSTEAD OF HAVING TICK MARKS FOR EACH INTEGER, THE FIGURE NOW HAS TICK MARKS ONLY FOR EACH BIN IN THE HISTOGRAM. EACH BIN SPANS THE TICK MARKS AROUND IT. TO MY EYE, THIS IS A STANDARD WAY OF PRESENTING A HISTOGRAM.

3. Lines 404-5 state: "Ongoing qualitative work with mentors, however, indicates that is not the case." There is no substantiation provided. If there are publications of this work, please cite them; if there are novel data, please present them (along with the methods). Otherwise, it should be restated that the authors believe it is not the case based on their anecdotal observations.

THIS HAS BEEN RESTATED AS ANECDOTAL EVIDENCE. THE QUALITATIVE WORK IS CURRENTLY BEING WRITTEN UP FOR PUBLICATION, SO HAS NO FORMAL CITATION AS YET

Reviewer #2: I am happy with the revision made by the author.

As I said in my previous review, the manuscript would benefit from a more elaborate discussion, comparing the obtained results with the prior literature. The author did not make this improvement, but the manuscript quality is sufficient for publication.

 I HAVE INCLUDED AN EXPANDED DISCUSSION OF THE IMPACTS THIS FINDING SHOULD HAVE ON THE STUDY OF OSS AND MENTORING IN THE MAIN BODY OF THE DISCUSSION SECTION, BEFORE THREATS TO VALIDITY

Thank you for your time!

Curtis Atkisson

---

## [Decision Letter · Decision Letter 2]

27 Jul 2022

Mentors matter: Association of mentors with project success in the Apache Software Foundation Incubator

PONE-D-21-32481R2

Dear Dr. Atkisson,

We’re pleased to inform you that your manuscript has been judged scientifically suitable for publication and will be formally accepted for publication once it meets all outstanding technical requirements.

Kind regards,

Maurizio Naldi

Academic Editor

PLOS ONE

Additional Editor Comments (optional):

Reviewers' comments:

Reviewer's Responses to Questions

**Comments to the Author**

1. If the authors have adequately addressed your comments raised in a previous round of review and you feel that this manuscript is now acceptable for publication, you may indicate that here to bypass the “Comments to the Author” section, enter your conflict of interest statement in the “Confidential to Editor” section, and submit your "Accept" recommendation.

Reviewer #1: All comments have been addressed

Reviewer #2: All comments have been addressed

2. Is the manuscript technically sound, and do the data support the conclusions?

Reviewer #1: Yes

Reviewer #2: Yes

3. Has the statistical analysis been performed appropriately and rigorously? 

Reviewer #1: Yes

Reviewer #2: Yes

4. Have the authors made all data underlying the findings in their manuscript fully available?

Reviewer #1: Yes

Reviewer #2: Yes

5. Is the manuscript presented in an intelligible fashion and written in standard English?

Reviewer #1: Yes

Reviewer #2: Yes

6. Review Comments to the Author

Reviewer #1: Thank you for your revisions. After reviewing your latest version of the manuscript, I am satisfied that all my previous critiques have been sufficiently addressed.

Reviewer #2: I am happy with the revision made by the author. In my opinion, the manuscript complies with all acceptance criteria now.

7. PLOS authors have the option to publish the peer review history of their article (what does this mean?). If published, this will include your full peer review and any attached files.

Reviewer #1: **Yes: **Kevin Buffardi

Reviewer #2: No

---

## [Editor Report · Acceptance letter]

9 Aug 2022

PONE-D-21-32481R2 

Mentors matter: Association of mentors with project success in the Apache Software Foundation Incubator 

Dear Dr. Atkisson:

I'm pleased to inform you that your manuscript has been deemed suitable for publication in PLOS ONE. Congratulations! Your manuscript is now with our production department. 

Kind regards, 

on behalf of

Professor Maurizio Naldi 

Academic Editor

PLOS ONE